# Uttarakhand State Earthquake Early Warning System: A Case Study of the Himalayan Environment

**DOI:** 10.3390/s24113272

**Published:** 2024-05-21

**Authors:** Pankaj Kumar, Mukat Lal Sharma, Ravi Sankar Jakka

**Affiliations:** 1Centre of Excellence in Disaster Mitigation & Management, Indian Institute of Technology Roorkee, Roorkee 247667, India; pkumar@dm.iitr.ac.in; 2Department of Earth Sciences, Indian Institute of Technology Roorkee, Roorkee 247667, India; 3Department of Earthquake Engineering, Indian Institute of Technology Roorkee, Roorkee 247667, India; m.sharma@eq.iitr.ac.in (M.L.S.); ravi.jakka@eq.iitr.ac.in (R.S.J.); 4Department of Mathematics, Indian Institute of Technology Roorkee, Roorkee 247667, India; pratibha@ma.iitr.ac.in

**Keywords:** Himalaya, Uttarakhand, UEEWS, instrumentation, lead time, mitigation tool

## Abstract

The increased seismic activity observed in the Himalayas, coupled with the expanding urbanization of the surrounding areas in northern India, poses significant risks to both human lives and property. Developing an earthquake early warning system in the region could help in alleviating these risks, especially benefiting cities and towns in mountainous and foothill regions close to potential earthquake epicenters. To address this concern, the government and the science and engineering community collaborated to establish the Uttarakhand State Earthquake Early Warning System (UEEWS). The government of Uttarakhand successfully launched this full-fledged operational system to the public on 4 August 2021. The UEEWS includes an array of 170 accelerometers installed in the seismogenic areas of the Uttarakhand. Ground motion data from these sensors are transmitted to the central server through the dedicated private telecommunication network 24 hours a day, seven days a week. This system is designed to issue warnings for moderate to high-magnitude earthquakes via a mobile app freely available for smartphone users and by blowing sirens units installed in the buildings earmarked by the government. The UEEWS has successfully issued alerts for light earthquakes that have occurred in the instrumented region and warnings for moderate earthquakes that have triggered in the vicinity of the instrumented area. This paper provides an overview of the design of the UEEWS, details of instrumentation, adaptation of attributes and their relation to earthquake parameters, operational flow of the system, and information about dissemination of warnings to the public.

## 1. Introduction

The economic repercussions of disasters have significantly increased manifold and are projected to grow continuously in the coming years [1]. In 2012, the climate-related financial deficit was 1% of the gross domestic product (GDP) of developing countries [2]. Moreover, losses attributed to seismic activities are also on an upward trajectory. The Himalayan region has experienced numerous devastating earthquakes in the past, and many experts predict their recurrence in the future [3,4,5,6,7,8]. Although the occurrence of earthquakes cannot be prevented, their impact on society can be reduced through the application of innovative technological solutions.

The inherent unpredictability of earthquakes renders them more perilous than other disasters. Seismologists have scrutinized many precursors to predict earthquakes, yet none achieved full confidence in all scenarios. Hence, a novel method emerged known as the Earthquake Early Warning (EEW) system. This system serves as a tool to mitigate the risks associated with earthquakes by providing early warnings to the users, potentially saving lives. Fully developed EEW systems worldwide have demonstrated their effectiveness in reducing casualties during earthquake events [9,10,11]. The EEW system is a live earthquake monitoring system capable of detecting the onset of an earthquake, estimating its probable magnitude, and issuing warnings before substantial ground shaking reaches the users’ locations [12]. Its primary focus is on providing timely alerts with a sufficient lead time required to take precautionary measures and shut down key facilities, rather than precisely determining earthquake parameters.

The fundamental principle of an EEW system relies on the differing propagation speeds of primary (also known as longitudinal, non-damaging, P-wave) and secondary (also known as shear, transverse, damaging, S-wave) waves. These waves are produced because of stress release during earthquakes. S-waves inflict notable damage as they travel roughly half the speed of P-waves and are considerably slower than electromagnetic signals. Telecommunication operates at the speed of electromagnetic signals, akin to the speed of light. Therefore, a system can be developed to take advantage of the speed of telecommunication to fetch data from sensors to a server and subsequently, perform analysis and issue warnings before damaging waves reach the users’ locations. The server comprises an assembly of high-performance computers for computational tasks and other telecommunication equipment such as switches and routers. The modern-era telecommunication and data transmission facilities between the sensors and server instill confidence in the efficacy of an EEW system. EEW algorithms operate on real-time data and decision-making modules issue warnings based on predefined threshold parameters [13]. It is preferable to establish precise threshold values of EEW parameters to obtain accurate, reliable, and fast results [14]. Lead time or warning time depends on factors such as the epicentral distance from the target locations and the geographical distribution of the stations around the epicenter. 

Initially, an EEW system was developed for the Tohoku Shinkansen railway system in Japan in 1982 by employing the front detection technique [15]. Mexico’s seismic alert system in 1991 also adopted a similar approach [16]. The Urgent Earthquake Detection and Alarm System (UrEDAS), which employs an improved front detection technique, commenced operations for the Tokaido Shinkansen line in 1992, utilizing the initial three seconds of P-wave motion following the P-wave onset [17]. After observing the successes of the EEW systems in Mexico and Japan, many other countries embarked on endeavors to develop EEW system customized to their respective regions. Significant progress has been evident in this field over the past three decades. EEW systems are operational in specific countries like Japan [18,19], Taiwan [20,21,22], Mexico [9,23,24], and South Korea [25], issuing nationwide warnings that cover a significant portion of the public. However, some countries issue region-specific warnings due to localized seismic vulnerability in certain areas, such as Anatolia in Turkey [26,27], Southwest Iberia [28,29,30], Southern Italy [31,32,33], Vrancea in Romania [34,35,36], Beijing in China [37,38], Chile [39], Costa Rica [40,41], Switzerland [42], Nicaragua [43], Israel [44,45], and the West Coast of the United States of America [11,46]. The advancement of EEW systems is pivotal for reducing seismic risk however enhancing the capacity to provide more lead time is equally essential [12]. An EEW system’s effectiveness in risk reduction relies on how quickly information is provided to the users, enabling them to have more lead time to react. This can be achieved by developing improved algorithms, estimating EEW attributes, calculating earthquake metadata, and issuing warnings rapidly.

## 2. Seismic Activity in the Himalayas and the Region of Interest

The Himalayas stand as one of the world’s most seismically active regions. Ongoing convergence over the past roughly 50 million years has led to the formation of numerous complex tectonic clusters in the region. The collision of the continental plate boundaries, specifically the Indian and Eurasian plates, led to the subduction of the Indian plate beneath the Eurasian plate [47,48]. Following the collision period, the Indian tectonic plate has been steadily advancing northward at an average rate of about 50 mm per year [49]. As a result, deformation occurred in the southward direction leading to the formation of faults, folds, and notable structural characteristics within the Himalayan orogenic belt [50]. The significant tectonic features include the Main Frontal Thrust (MFT), Main Central Thrust (MCT), Main Boundary Thrust (MBT), and the Indus-Tsangpo Suture Zone (ITSZ) [51,52,53]. The ITSZ dips southward, whereas MCT and MBT dip northward [54]. The MCT was active during the earlier period and is considered the oldest thrust system. Conversely, the MBT currently displays an active thrust system. The MFT represents the southernmost and youngest thrust system. The movement of tectonic plates induces strain along fault plate boundaries, resulting in the accumulation of stress [55]. Earthquakes occur as a result of the catastrophic failure of rocks and the sudden release of accumulated stress [56].

The geodynamic activity in the Himalayas has resulted in numerous devastating earthquakes, with one of the most recent being the Nepal earthquake of 25 April 2015, of a magnitude *M_w_* 7.6, along with its subsequent aftershocks, which caused substantial economic loss to the Himalayan country. Following this earthquake, the Nepalese government conducted a post-disaster need assessment, estimating the total direct and indirect economic losses to be close to USD 7 billion, equivalent to approximately one-third of the country’s GDP in the year [57]. Among the notable past earthquakes in the Himalayas is the Kangra Valley earthquake of 4 April 1905, of a magnitude *M_s_* 7.8. This event marked the first of several devastating earthquakes of the 20th century. According to estimates by the then Punjab government, that earthquake resulted in approximately 20,000 casualties among the population of 375,000. The economic cost of recovery due to this earthquake was estimated to be around 2.9 million rupees in 1905 [58].

Similarly, the 12 June 1897, Great Assam earthquake of a magnitude *M_w_* 8.0 was very devastating, and its tremors were felt up to the Peshawar region (now in Pakistan). The then British government surveyed post-earthquake damage and stated in the reports that the infrastructure of the eastern region was majorly devastated [59]. The 15 August 1950 Tibet-Assam Great earthquake of magnitude *M_w_* 8.6 shocked the eastern Himalayan region [60] and caused 1526 casualties and extreme economic loss of around USD 25 million at that time [61]. The 15 January 1934 Bihar-Nepal earthquake of a magnitude *M_w_* 8.0 rattled the border region, and tremors were felt over an area of approximately 4,920,000 km^2^ in India, Nepal, and Tibet [62]. That seismic event resulted in significant destruction, generating numerous fractures and triggering landslides [63], killing approximately 12,000 people [64]. In the 1990s, two strong earthquakes rattled the Garhwal region of Uttarakhand. The Uttarkashi earthquake of *M_w_* 6.8 on 19 October 1991, and the Chamoli earthquake of *M_w_* 6.6 on 28 March 1999, caused significant losses in the Uttarakhand region [65,66,67]. Sharma (2003) estimated the return periods of moderate to great earthquakes, with the conclusions aligning with other studies [68,69,70,71], expressing the increased frequency of occurrence and postulating urgent need and immediate attention to mitigation measures [72,73,74,75]. The epicenter of the 2015 Nepal earthquake was approximately 450 km from the Uttarakhand region. Considering that Uttarakhand shares similar seismological conditions with Nepal, the occurrence of an earthquake of similar magnitude in Uttarakhand could have devastating consequences. Hence, a lot of efforts are being made to curtail exposure and vulnerability in anticipation of future earthquakes in Uttarakhand.

Since the 1950 Assam earthquake, no great earthquake (*M_w_* 8 or above) has occurred in the Himalayas. Srivastava et al. (2015) analyzed Himalayan seismicity by examining seismic patterns, local tectonics, global positioning system (GPS) observations, microearthquakes, paleo seismicity, and other pertinent datasets. They identified two distinct types of seismic gaps with unique characteristics [71]. Category-1 seismic gaps are found in regions where significant earthquakes (*M_w_* ≥ 8) have either occurred historically or are anticipated in the future. Specifically, category-1 gaps include the Kashmir seismic gap, West Himachal-Pradesh seismic gap, Uttarakhand-Dharchula seismic gap, Central Nepal-Bihar seismic gap, Arunachal seismic gap, and Shillong seismic gap. Category-2 gaps delineate regions where significant earthquakes (*M_w_* < 8) have either occurred historically or are expected to occur in the future. These include the Jammu, East Himachal-Pradesh, Western Nepal, and Sikkim-Bhutan seismic gaps. The region situated between the 1905 Kangra earthquake and the 1934 Bihar-Nepal earthquake has been identified as the central seismic gap [76]. GPS measurements conducted in the Himalayan regions indicate accumulation of strain in this area, possibly resulting in one or more significant earthquakes [77,78]. After a thorough examination of the intricacies involved, the utilization of constant seismicity and constant moment rate methods, along with time-dependent occurrence models, has revealed return periods of earthquakes with different magnitude ranges in the Himalayas [79,80,81,82].

The Uttarakhand-Dharchula seismic gap lies within the central seismic gap and has not witnessed noteworthy major earthquakes in recent recorded history. This seismic gap comprises an approximately 800 km long central segment commonly known as the Garhwal-Kumaon Himalaya. It is often described as an uninterrupted section of the Himalayan arc [69]. However, the Himalayan region is experiencing swift urban expansion, rapid settlement, and significant infrastructure development [83,84,85]. According to the 2011 census, the Indian Himalayan region spanning Kashmir in the north to Arunachal Pradesh in the east is home to around 52.8 million people [86]. With an annual growth rate of 3.30%, the population is projected to surpass 260 million by the end of 2061 [87]. In the event of a major earthquake occurring in the central seismic gap region, the potential for thousands of casualties and substantial economic losses amounting to billions of dollars [88] underscores the urgent need for implementing mitigation measures and disaster risk reduction strategies in this area. Recognizing the heightened seismic risk and dispersed vulnerability within the central seismic gap, an earthquake early warning (EEW) system for this region was proposed. Initially, a two-pronged approach was adopted. First, a feasibility study was performed about the available emerging solutions. Subsequently, the decision was made to establish the first Indian regional EEW system, UEEWS. This paper outlines the development phases of the system, including instances of successful notifications, alerts, and warnings issued.

## 3. Architecture of the Developed UEEWS

The UEEWS was developed and proposed to address the specific needs of Uttarakhand in central Himalaya after a comprehensive study of the existing EEW systems present in the world [89,90,91,92]. Due to high seismicity and the pressing need for mitigation measures, the central seismic gap region was chosen for establishing India’s first EEW system. Subsequently, a cost-benefit analysis was carried out, leading to the finalization of sensor selection [89]. For this, the available strong ground motion recording sensors underwent thorough scrutiny before the procurement process. The cost was a constraining factor, compounded by the significant number of sensors needed for establishing a regional earthquake early warning system. Consequently, low-cost micro-electromechanical systems (MEMS)-based sensors were finalized. The developed system follows the regional early warning approach; hence, a control room was set up at the Earthquake Early Warning System (EEWS) Laboratory in the Centre of Excellence in Disaster Mitigation & Management (CoEDMM), Indian Institute of Technology (IIT) Roorkee.

### 3.1. Seismic Network

During the initial instrumentation phase, the Garhwal region in the central Himalayas, spanning from Joshimath in the East (30.5616° N, 79.5594° E) to Mori in the West (31.0183° N, 78.0409° E) was selected. This area encompasses approximately 150 by 50 km^2^ across five districts of Uttarakhand: Uttarkashi, Tehri, Rudraprayag, Pauri and Chamoli. As a dense network is preferred for a regional earthquake early warning system; therefore, low-cost MEMS-based sensors were selected. These sensors offer a relatively lower dynamic range compared with conventional high-cost force balance accelerometers (FBA) with larger dynamic range. The MEMS-based accelerometer (i.e., pAlert) fulfills the requirement of regional EEW systems and is cost-effective compared with conventional FBAs. These sensors comprise tri-axial MEMS accelerometers paired with a 16-bit 80 MHz CPU, offering an output resolution of 16 bits and a dynamic range exceeding 86 dB. The required EEW parameter (low-pass filtered vertical displacement, *P_d_*) is efficiently calculated from the data retrieved from these sensors. These sensors have undergone testing and have also been successfully deployed in Taiwan’s EEW system [93,94].

The pAlert can transmit data to two servers over the internet by utilizing the pAlert-to-Earthworm module (PALERT2EW) running on the servers and automatically synchronizing its time through the network time protocol (NTP). PC-Utility 1.14 software is utilized for remote access to the pAlert and configuration purposes. In the initial phase, instrumentation of pAlert for the UEEWS was completed as a pilot project from 2014 to mid-2017. Figure 1 shows the location of 84 sensors installed in the first phase. These sensors are mounted on the ground floor of government-owned offices of the base transceiver station (BTS) of Bharat Sanchar Nigam Limited (BSNL) and the point of presence (PoP) of the state-wide area network (SWAN) available in the Garhwal region of Uttarakhand. 

To cover the remaining portion of the current seismogenic source, the instrumentation was expanded in mid-2017 to include the Kumaon region of Uttarakhand. The region encompasses four districts: Bageshwar, Pithoragarh, Champawat and Nainital. For this expansion, an upgraded version of the sensor (pAlert+) was chosen. These sensors feature 24-bit tri-axial MEMS accelerometers with an internal memory of 8 GB and dynamic range exceeding 100 dB. The sensors are mounted on the ground floor of BSNL’s BTS and the SWAN’s PoP in the Kumaon region. As shown in Figure 1, an additional 86 sensors are installed in the Kumaon region, bringing the total to 170 sensors in the entirety of Uttarakhand. The inter-station spacing ranges from about 10 to 20 km. Consequently, the network now satisfies both criteria of the regional EEWS: the necessary sensor density and the spatial coverage of the seismogenic source in Uttarakhand, enabling the detection of large earthquakes in the instrumented region.

The logical structure for established telecommunication from sensors to the control room is shown in Figure 2. In this figure, the sensors installed in the field are accelerometers, connected by a communication device-an asymmetric digital subscriber line (ADSL) or an optical terminal line (ONT) as per the availability and are connected to the optical line terminal (OLT) at the stations. Networking of the sensors installed in the field passes through different communication media lines like the BSNL cloud, multiprotocol label switching (MPLS) networking, then the OFC network from Haridwar to Roorkee to the installed router, switch, server, and multiple PCs for numerous operations in the laboratory.

### 3.2. Data Streaming

The decision to install sensors in BSNL’s BTS and the SWAN’s PoP buildings was driven by the provision of power availability, a dedicated virtual private network over broadband (VPNoBB), and the necessary security required for the sensor as well as associated accessories. All the sensors are installed on the ground floors of these buildings, transmitting ground-motion data to the server located in the EEWS Laboratory. This network operates continuously in real time, 24 hours a day, 7 days a week. Figure 3 illustrates the streaming of data from sensors to servers in real time using Swarm 3.2.0 software. The installed sensors transmit data every second, packaged into 1200-byte packets. As all sensors within the network synchronize their clocks with the UEEWS server, the NTP server adjusts each sensor’s time accordingly.

### 3.3. Data Processing

As the UEEWS marks the first EEW system developed for a Himalayan region, efforts have therefore been made to employ open-source software to facilitate easy adoption in other Himalayan areas. Earthworm 7.10, an open-source platform used to monitor earthquakes and volcanoes and process seismic data [95,96], serves as a module-based seismic network processing platform. Its open-source nature allows the customization of modules to meet specific requirements. Leveraging the flexibility of Earthworm, the central server is set up with the customized existing modules and incorporating new modules designed to suit our earthquake parameter estimation needs.

A two-pole Butterworth high-pass filter with a cut-off frequency of 0.075 Hz is applied over the streamed data in real time to remove low-frequency noises. The PALERT2EW module is used to fetch the streamed data into the shared memory called WAVE_RING, while the WAVE_SERVERV module is responsible for storing and providing seismic waveforms over time, displaying streamed data. Figure 4 illustrates the data flow from the field to the central server. Swarm 3.2.0, another open-source software, plots the accelerograms of the data, offers real-time streaming of data from installed sensors, and supports historical data analysis.

### 3.4. Event Detection

Accurately identifying the onset of the P-phase within continuously streaming seismic signals is crucial for the effectiveness of an EEW system. Many algorithms have been developed in the past for detecting the P-phase onset in seismic signals [97,98,99,100,101,102,103,104,105,106]. The UEEWS employs an enhanced version of the standard short-term average (STA) and long-term average (LTA) algorithm [21,107,108].

The PICK_EEW module is deployed to detect P-phase arrivals in the continuously streaming vertical channel data. This module also estimates peak amplitudes of P-wave displacement (*P_d_*), velocity (*P_v_*), and acceleration (*P_a_*) from 3-s time-windowed data following P-onset, then forwards this information to another shared memory named PICK_RING. The PICK_EEW module [21] is an improved version of the Earthworm’s default PICK_EW module. The PICK_EEW module applies the standard STA-LTA algorithm [107,108].

Due to potential variations in noise levels caused by natural surroundings and human activities at different installation sites, the PICK_EEW module incorporates two additional parameters, *P_a_* and *P_v_*, to mitigate false detections triggered by surrounding disturbances [21]. This module reads a configuration file containing the threshold values of the parameters to obtain a true pick of the P-onset. The parameter values vary for each station, so the configuration file contains their respective values. Thus, site-specific adjustments to the configuration file have been made based on the analysis of earthquake data collected since 2014, the year of commencement of this project. This involved replaying recorded data on an offline server to facilitate improvements in the configuration file for each station.

### 3.5. Estimation of Earthquake Parameters

A module named TCPD was created in the C programming language to estimate earthquake parameters, designated to integrate seamlessly with the Earthworm platform [96]. This module underwent modifications to align with the requirements of the UEEWS. This module accesses the PICK_RING and performs analysis to estimate earthquake parameters including location, magnitude, depth, and origin time. Subsequently, it updates the information obtained from this ring and transfers it to another memory location referred to as the EEW_RING.

#### 3.5.1. Location Estimation

The process of estimation of the earthquake hypocenter involves two main steps. First, the Geiger method, an inversion process, is employed to determine the epicenter [109]. This method requires a P-wave velocity profile specific to the instrumented region. For this purpose, the half-space velocity model recommended by a study conducted for the Uttarakhand region has been adopted [110]. In the second step, the grid search method is utilized to ascertain the actual depth of the earthquakes, ranging from 0 to 50 km. Although alternative methods like the back azimuth method employing principal component analysis are advantageous for locating long-distance earthquakes, they are deemed unsuitable for near-source recordings [111]. In the grid search method, the theoretical travel time to each triggered station is computed and compared with the observed depth at each iteration. The depth of the earthquake is determined by identifying the step at which the minimum residual is encountered. Initially, at least four triggered stations are used, and the search point that yields the minimum residual is selected as the focal point. For the UEEWS, the search region is confined within a radius of 200 km from the position of the initial trigger with a depth of 50 km, ensuring that the search area remains within the array. However, in the case of earthquakes originating from outside the instrumented area, the search region is extended beyond the instrumented area, but within 200 km from the location of the first trigger. The limit of 200 km was set up after considering the area of the instrumentation. The expansion of instrumentation spans 280 km east-west and 130 km north-south. If an earthquake occurs in the border area, its vibrations will spread in all directions and will be recorded by sensors installed in the border area. With the help of this recorded data UEEWS server will calculate the location and other parameters of the earthquake.

This approach provides an epicenter’s accuracy of approximately 5 km within the search region. Hence, precise locations can be determined for earthquakes originating outside but in close proximity to the instrumented region. Its capability to achieve an accurate location with approximately 5 km precision in focal depth, coupled with its reduced computational requirements compared with classical methods, makes it well-suited for the EEW system [21].

This entire procedure is encapsulated within a function in the TCPD module, which is invoked when there are P-phase picks from at least four stations. Upon meeting this condition, the TCPD module initiates the process to locate the hypocenter. If the estimated root mean square (RMS) of travel-time residuals obtained from the inversion process surpasses the specified threshold, the pick with the highest travel-time residual is discarded. Meanwhile, if a new pick is identified, the procedure to locate the hypocenter restarts again and updates the previously estimated hypocenter.

#### 3.5.2. Magnitude Estimation

Once the hypocenter is determined within the TCPD module, another function is invoked to estimate the magnitude. This function utilizes a regression model to calculate the magnitude of earthquakes. The mathematical expression of the employed model is as follows:*M* = A × log (*P_d_*) + B × log(*R*) + C(1)

In this equation, *R* represents the hypocentral distance, calculated as the square root of the sum of the squares of epicentral distance (*d*) and focal depth (*h*), and is estimated as *R* = √(*d*^2^ + *h*^2^). *P_d_* is the estimated peak displacement using three seconds of data after P-onset. A, B, and C are coefficients in Equation (1) and vary depending on the dataset used as well as the region.

Initially, Wu and Zhao (2006) utilized this model to estimate *P_d_*-based magnitude using Southern California Seismic Network (SCSN) data in the following manner [112]:*M_Pd_* = 4.748 + 1.371 × log(*P_d_*) + 1.883 × log(*R*)(2)

A new model was devised to calculate magnitude based on the peak displacement (*P_d_*) of the P-wave’s vertical component, utilizing 70 records of 13 earthquakes that transpired in the Uttarakhand region between 2005 to 2020. The attenuation relationship of *P_d_* with hypocentral distance (*R*) and magnitude (*M_w_*) is given below (Equation (3)).
log(*P_d_*) = −2.6826 + 0.52258 × *M_w_* − 1.2011 × log(*R*) ± 0.252 (3)

The dataset used in the development of this new model is limited and includes records of 3.6 < *M_w_* < 5.5 with 15 < *R* < 100. Thus, the model is a good fit for small earthquakes (3 < *M_w_* < 5). After inverting this new relationship, *M_Pd_* can be estimated, and it has a 1:1 relationship with catalog magnitude (*M_w_*).

Due to the insufficient number of large earthquakes recorded in the Indian dataset for the Uttarakhand region, the model proposed by Hsiao et al. (2011) is currently being utilized in the UEEWS to estimate *P_d_*-based magnitude (Equation (4)) [113].
*M_Pd_* = 3.905 + 2.198 × log(*P_d_*)+ 2.703 × log(*R*)(4)

#### 3.5.3. Report Generation

The TCPD module generates a report and stores it in a location specified by the server administrator for archival purposes. Once the report is generated, a warning is promptly issued to the public. A threshold of magnitude five has been set for issuing warnings. The earthquake report contains essential parameters such as latitude and longitude of the epicenter, depth, origin time, list of triggered stations, P-wave arrival time, etc. A decision-making flowchart for issuing warnings is depicted in Figure 5. The parameters of small earthquakes detected by the UEEWS are outlined in Table A1. For comparison purposes, the source parameters for the same earthquakes, as reported by the National Center for Seismology (NCS), Government of India, are also provided in Table A2.

## 4. Warning Modes

Warnings to the public can be disseminated through multiple modes, including application-specific sirens, television broadcasts, AM/FM radio, mobile messages, and mobile app. In the present setup of the UEEWS, individuals receive alerts by two modes: sirens and mobile app. Sirens are strategically installed at government-owned buildings, schools, hospitals, and residential complexes, while the mobile app is publicly available to download in smartphones.

### 4.1. Sirens

The EEWS Laboratory indigenously developed the siren units. The design of the UEEWS siren is divided into four main modules [114]. First, the controller board manages the siren’s operation and stores various warning sounds. The second component consists of an amplifier circuit, which boosts the sound signal from the controller board. Following this is the speaker/hooter, which converts these electrical signals into audible sounds. Lastly, the power supply unit converts 230 V AC into DC, necessary for powering the controller and amplifier. Figure 6 depicts a flowchart outlining these modules.

The controller board functions as a microcomputer, akin to a Raspberry Pi, linked to the internet through a LAN port. It includes general-purpose input-output (GPIO) pins for interfacing with other devices. A computer program communicates with the warning server over the internet to receive real-time warning messages and manages hooter/speaker relays via GPIO pins. Figure 7 illustrates the logic diagram outlining the functionality of the computer program.

Appendix A illustrates the installation of sirens across various locations, including the State Emergency Operation Center (SEOC) in Dehradun, the capital city of Uttarakhand, all District Emergency Operation Centers (DEOCs) within the state, as well as key government buildings in the two major cities of the state, Dehradun and Haldwani. Appendix A is given in the Appendix A.

### 4.2. Mobile Application

The Uttarakhand government, in collaboration with IIT Roorkee, has installed public sirens in two major cities, Dehradun and Haldwani, as well as at various DEOCs. However, due to the extensive area and large population, a considerable number of sirens would be needed to cover Uttarakhand, requiring significant investment of both funds and time. This prompted us to explore alternative options to enhance the effectiveness of the developed system. In today’s digital world, Android and iOS-based smartphones are exceedingly prevalent and owned by a significant portion of the population. Therefore, it was deemed important to issue warning messages on mobile app also. The developed app should include some unique features like loud distinctive sound along with voice message when warning is received, so that users get cautious immediately. The app should also provide guidance on actions to be taken in the event of an earthquake and instructions on identifying safe areas within the home for taking cover. Additionally, the app should feature a graphical interface displaying a visual representation of the remaining time before the S-wave reaches the user’s location.

Therefore, IIT Roorkee has developed a smartphone app called BhuDEV (Bhukamp Disaster Early Vigilantè) to disseminate earthquake early warnings, with the aim of alerting users so that they could take precautionary measures before the arrival of damaging waves at their locations. To receive such warnings, users need to install the app and provide essential details during the registration process. The app also features instructional videos for users to take proper safety steps for themselves as well as their loved ones. Currently, the app provides early warnings for damaging earthquakes originating exclusively in Uttarakhand.

When the server anticipates an imminent earthquake with a magnitude of five or above, public warnings are issued indicating potential damage. Similarly, notifications are sent out for earthquakes of lower magnitude, below five. This app needs the current location of smartphone users to dispatch SMS alerts to the registered contacts on press of SOS button to share their present location with the Uttarakhand State Disaster Management Authority (USDMA) during earthquake emergencies. Consequently, permission for location access should be given during app installation and maintained continuously. Hence, granting access to location sharing is essential. This sharing of location assists search and rescue teams in promptly commencing operations during post-earthquake operations. The app receives warnings via the internet, requiring users to always maintain an internet connection. However, data usage is limited to earthquake notifications and warnings only. Additionally, the app helps in simulating earthquake scenarios during mock exercises and provides video links for preparedness and better comprehension of earthquakes.

On 4 August 2021, the honorable Chief Minister of Uttarakhand officially launched this app to the public. The app can be downloaded either by scanning the QR code or by accessing it from the Play Store or App Store. The app icon is provided in Appendix A for easy identification, and a few pages of the mobile app after installation are showcased in the Appendix A.

## 5. System Application Effectiveness—The Case of the Tehri Garhwal Earthquake Warning

Lead time denotes the duration of warning time offered to the users during earthquake events. It indicates how long it takes for S-waves to reach a specified location, which varies depending on the distance of human settlements from the epicenter of the earthquakes. Settlements at far distances get more lead time compared with those nearest to the epicenter. The current system operates based on the regional EEWS concept, requiring a few seconds to make decisions. These seconds include the total time required to gather three seconds of data from at least four sensors, and time lapsed in the following processes: transmission of data to the server, their processing, estimating magnitude and hypocenter, and subsequently relaying the decision to the warning server for issuing warnings to the public.

The lead time (*Tw*) can be calculated using the formula: *Tw* = *Ts* − *Tr*. Here, *Tr* represents the reporting time and is estimated as *Tr* = *Td* + *Tpr*, where *Td* signifies the time required to trigger and record an adequate length of waveforms, and *Tpr* is the processing time needed to process the waveforms for determining the hypocenter and magnitude. *Ts* denotes the travel time of the damaging S-wave, and for the advanced warning, it is essential to have *Tw* greater than 0. This condition implies that *Ts* must exceed *Td* + *Tpr*. Settlements with *Tw* < 0 are categorized as blind zones, suggesting the necessity for an onsite warning system. The EEWS Laboratory is currently engaged in developing such a system. The lead time is further explained below, accompanied by a detailed example.

Figure 8 shows a diagram illustrating discrete times based on data from the Tehri Garhwal earthquake that occurred on 6 November 2022, at 03:03:03 UTC, of a magnitude *M_w_* 4.5 and a depth of 5 km. From the epicenter, the Maneri (MNRB) station was 10.27 km away, the Chamba (CMBS) station was 40.87 km away, and the Artola (ARTB) station was 166.19 km away. In this earthquake, the first P-onset was detected at the MNRB station, while the last P-onset was registered at the CMBS station during the initial report generated by the TCPD module. Consequently, the time required for triggering a sufficient number of stations and recording sufficient primary wave data (*Td*) was 7.81 s from the origin time. The processing time (*Tpr*) needed to calculate the hypocenter and magnitude was 4.38 s. Hence, there was a lead time (*Tw*) of approximately 38 s for the ARTB station before the arrival of destructive waves, which took approximately 50 s (*Ts*) from the origin time to reach this station.

The TCPD module promptly generates earthquake reports upon detecting P-onset signals from at least four stations. It initiates the earthquake parameter estimation process and generates a report accordingly. Upon report generation, the information is relayed to a warning server, which, based on the estimated magnitude, issues either alerts or notifications. Warnings are issued for earthquakes with a magnitude exceeding five, while notifications are dispatched for those below magnitude 5.0. If the TCPD module detects another new P-onset, it reinitiates the parameter estimation process, generates earthquake reports, and disseminates warnings/alerts to the public.

During the Tehri Garhwal earthquake on 6 November 2022, the TCPD module generated six reports. Seismic data from this earthquake were recorded at 96 stations. Post-processing was performed to estimate the timings of P-onset and S-onset from the accelerograms of each recording station. Following this, the time taken for the S-waves to reach these stations after issuance of notification was estimated. The lead time was then calculated based on the time difference between the issuance of warnings and the arrival of S-waves. Figure 9 illustrates the lead time for all functioning stations at that time. The pink oval indicates a blind zone area where no notifications could have been provided because the processing time by EEWS to estimate parameters and issue warnings was more and, in the meantime, S-waves had already crossed those locations.

Figure 10 displays a screenshot of the notifications received during this event on a user’s mobile app. Notifications issued to the public commence from the third generated report onwards, while for the initial two reports, UEEWS waits to confirm that they are indeed generated because of an earthquake. The user’s mobile time is shown in the hh:mm format without seconds. For sake of precise information, the timing of the notifications issued by the server is provided in the Indian Standard Time format. 

## 6. Performance of UEEWS

The dense instrumentation of the UEEWS ensures that any earthquake above magnitude 4.0 in the instrumented region would be recorded by at least four stations. On 8 February 2020, 01:01:50 UTC, a light earthquake with a magnitude *M_w_* 4.7 and a depth of 48.2 km struck Pithoragarh district of Uttarakhand. Figure 11 illustrates the epicenter and the triggered sensors’ locations, while Figure 12 displays the recorded accelerograms by the vertical channel of the sensors. An early notification was automatically issued solely to a close group of researchers since the system was not unveiled to the public at that time.

Subsequently, on 11 September 2021, a magnitude *M_w_* 4.7 earthquake struck in the Chamoli region. This event marked the first earthquake following the launch of the mobile app and the inauguration of the system to the public. Since it was a minor earthquake, alert messages were disseminated to the public. This event represents the first incident during which UEEWS successfully issued notifications to the public for the first time.

The UEEWS has produced reports for 19 earthquakes from 2015 to the present, as indicated in Table A1. Among these, 14 earthquakes were triggered in the Uttarakhand region, while 5 occurred in the Nepal region. In line with the UEEWS objectives, our analysis focuses solely on earthquakes triggered in the Uttarakhand region. The comparison of earthquake information (magnitude, location) estimated by the UEEWS and NCS is shown in Figure 13, revealing significant variation in the epicenter and depth. The location estimation section describes the procedure to estimate the epicenter and depth. The variations in hypocenter determination by the UEEWS and NCS were influenced in part by differing velocity models and phase-picking methods. Additionally, the UEEWS employs cost-effective MEMS-based accelerometers, whereas NCS utilizes high-end state-of-the-art broadband seismographs and strong motion accelerographs. Furthermore, the UEEWS relies on the real-time analysis of the initial segment of seismic records, while NCS utilizes the complete earthquake waveform and provides source parameters in ~5–10 min after the earthquake, with an average time of about 8.0 min [115].

The average disparities in the estimated depth and epicenter between the UEEWS and NCS were 11.1 ± 13.6 km and 24.5 ± 22.6 km, respectively. The response time of the system is calculated as the duration between the origin time and the moment when the initial report is generated on the server. The average response time based on the reports generated during fourteen earthquakes (Table A1) was estimated as 13.6 ± 3.8 s. In one event (23 May 2021), the average response time was 22.38 s and after excluding this event, the average response time came down to 12.8 s.

Magnitude estimates are derived from P-wave data, leading to significant uncertainties in magnitude determination. This discrepancy arose from the use of various regression equations (Equations (2)–(4)) at different times and their subsequent updates. The estimated *M_Pd_* from the UEEWS server and magnitude (*M_w_*) from the NCS catalog can be plotted in a 1:1 relationship (Figure 14). In UEEWS, the magnitude estimation relies solely upon the initial portion of the P-wave following the P-onset; therefore, the estimated magnitude exhibits variation compared with the magnitude based on the moment estimated by NCS (Figure 14a). The standard deviations between the estimated magnitudes of the initial and final reports from the UEEWS and the magnitudes in the NCS catalog were 0.76 and 0.58, respectively. The discrepancy in the determined earthquake source parameters fluctuates with the earthquake reports generated by the server and gets improved as more triggered sensors are detected. The earthquake records given in Table A2, underwent re-analysis on the offline server, utilizing the model outlined in Equation (4). It resulted in notable enhancements in the estimated magnitudes. Consequently, the standard deviation of the first and final reports compared with the NCS catalog magnitude improved to 0.47 and 0.30, respectively (Figure 14b).

## 7. Discussion

The developed UEEWS has successfully issued notification for light earthquakes triggered in the instrumented region of Uttarakhand. Since the launch of the mobile app, BhuDEV, no strong earthquake has triggered in the Uttarakhand region; hence, notifications were issued for only light earthquakes of magnitude below five. However, warnings were issued for three moderate and two strong earthquakes triggered in the Nepal region. Despite their epicenters being 100 to 200 km away from the instrumented region, seismic waves were recorded by the installed sensors in Uttarakhand. UEEWS server analyzed the received seismic data while PICK_EEW module identified P-onset. Subsequently, the TCPD module utilized the seismic data associated with these P-onsets to estimate the earthquake parameters and generated earthquake reports accordingly. Upon detection of a new P-onset from a different station, the TCPD module repeated the same process to estimate the earthquake parameters and subsequently issued new notifications and alerts based on the estimated magnitude.

However, the estimated parameters provided by UEEWS did not align closely with those estimated by other agencies such as NCS. This disparity arose from the fact that UEEWS bases its parameter estimations on primary waves recorded exclusively by MEMS-based sensors. In contrast, NCS utilizes seismograms recorded by the state-of-the-art broadband seismographs, which offer a high dynamic range for more accurate estimation of earthquake parameters. The low-cost MEMS-based sensors offer cost-effectiveness but have limited capacity to accurately estimate complex earthquake parameters. Nevertheless, these sensors effectively serve the intended purpose of a regional EEWS. 

The primary objective of the UEEWS project is to provide timely warnings to the public before the arrival of damaging waves. This objective is being achieved with the current instrumentation. However, due to the constraints of MEMS-based sensors, the expected results for several other EEW parameters cannot be produced consistently. Therefore, these parameters are omitted, and the EEWS relies primarily on peak displacement as the key EEW parameter.

## 8. Conclusions

In conclusion, the UEEWS stands as one of the pioneering initiatives in seismic hazard mitigation efforts in the Himalayan region. Leveraging insights from global EEW systems, the UEEWS has been meticulously crafted to address the unique seismic challenges of the region. The work on developing EEWS for Uttarakhand was started in 2014 under a pilot project entitled “Development of Earthquake Early Warning System for North India”. Under this project, instrumentation in the Garhwal region was started and completed by mid-2017. The sensors were mounted on concrete platforms built on the ground floors of the BSNL and SWAN buildings. These buildings were selected due to their affiliation with telecommunication services, provision of electricity, and assurance of security for the installed sensor assemblies. A server has been set up in the EEWS Laboratory at CoEDMM, IIT Roorkee, along with other supporting communication devices. The network is maintained as private for security purposes. Seismic data is continuously streamed to the central server via dedicated lease lines in real time, having latency in milliseconds only. Different modules are executed on the open-source platform, Earthworm. After the successful development of the pilot project, the complete system was taken over by the Government of Uttarakhand and its instrumentation was extended into the Kumaon region; thus, a total of 170 sensors have been deployed across Uttarakhand under UEEWS. Despite the rugged terrain of the Himalayas posing challenges to internet connectivity for streaming ground motion data, the system remains under constant surveillance to ensure its operational integrity.

This system is currently full-fledged operational, issuing warnings to the public through blowing installed sirens and sending warnings on the mobile app installed in the smartphones by the users. This mobile app provides an SOS button useful for users during earthquakes. Upon pressing this button, the user’s location is transmitted to the disaster management authority and two registered relatives. The mobile numbers of relatives are entered during the mobile app registration process. Upon receiving the location information of the earthquake victim, the disaster management authority initiates search and rescue operations swiftly. The developed system serves not only to issue warnings to the public but also to establish a robust ground motion database, proving invaluable for earthquake and civil engineering applications [116]. This database can be utilized for analyzing seismic hazards in the region and formulating new ground motion prediction equations [117,118,119,120]. The developed system offers lead times ranging from seconds to tens of seconds to urban areas, towns, and rural villages within the state. One significant outcome of this initiative is the increased public awareness regarding natural hazards, especially earthquakes, as evidenced by the significant number of downloads of the mobile app. As a testament to its success, the system continues to evolve, striving for greater effectiveness and resilience in mitigating seismic risks and safeguarding communities in Uttarakhand.

## 9. Future Outlook

The present UEEWS relies on the VPNoBB service provided by BSNL and the SWAN network. In addition to this, communication may face disruptions due to various factors such as extreme weather conditions, landslides, damage to optical fiber cables, power outages, interruptions in data streaming, and so forth. Therefore, exploring alternative options such as public networks, cloud-based services, and solar power backups could prove advantageous.

Currently, the warning system does not provide information about the intensity of the earthquake at the user’s location. This feature could be incorporated once the prediction of strong ground motion and its conversion to intensity is integrated into the algorithm.At present, warnings are issued based on peak displacement (*P_d_*) of the first three seconds of P-wave data after P-onset from at least four sensors. However, there are various other attributes such as thepredominant period ($\tau_{i}^{p}$), characteristic period ($\tau_{c}$), cumulative absolute velocity (*CAV*), squared velocity integral (${IV}^{2}$), log averaged period ($\tau_{log}$), root sum square cumulative velocity (*RSSCV*), etc., may be explored in the future.Due to the intricate nature of Himalayan tectonics, it is recommended to deploy a dense network featuring wider aperture arrays.The issuance of warnings should also be vetted for their societal and management implications.

## Figures and Tables

**Figure 1 sensors-24-03272-f001:**
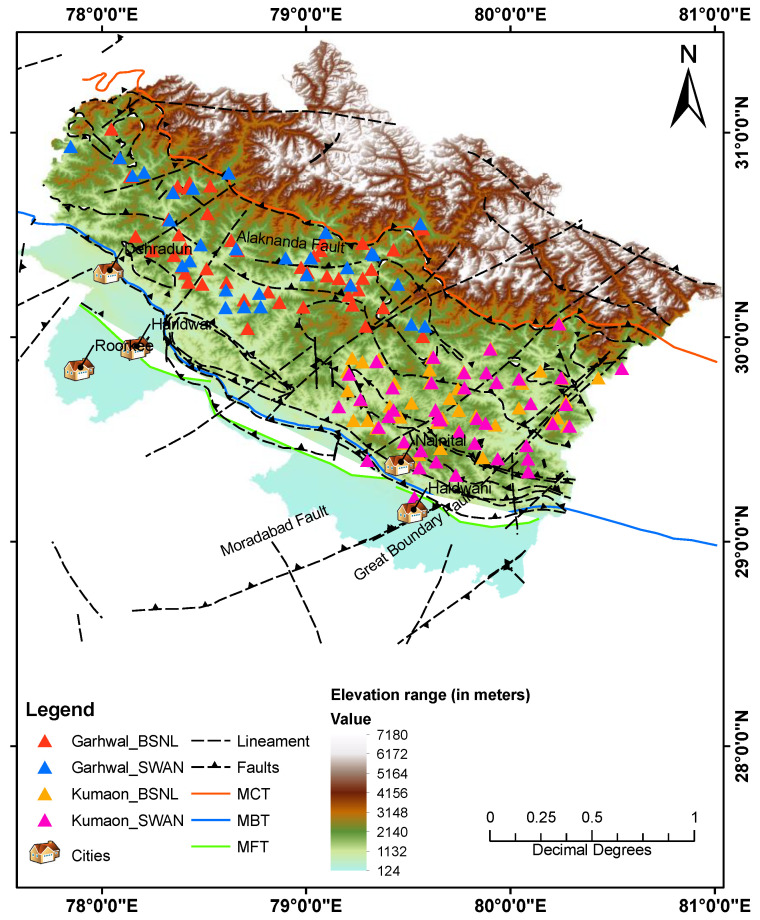
Location of the sensors installed in seismogenic areas of Uttarakhand.

**Figure 2 sensors-24-03272-f002:**
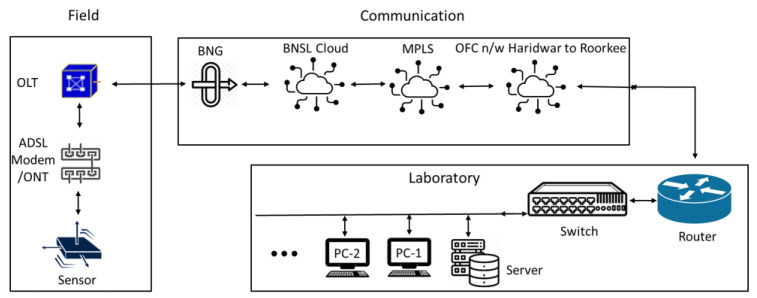
Logical structure diagram of the collection system network.

**Figure 3 sensors-24-03272-f003:**
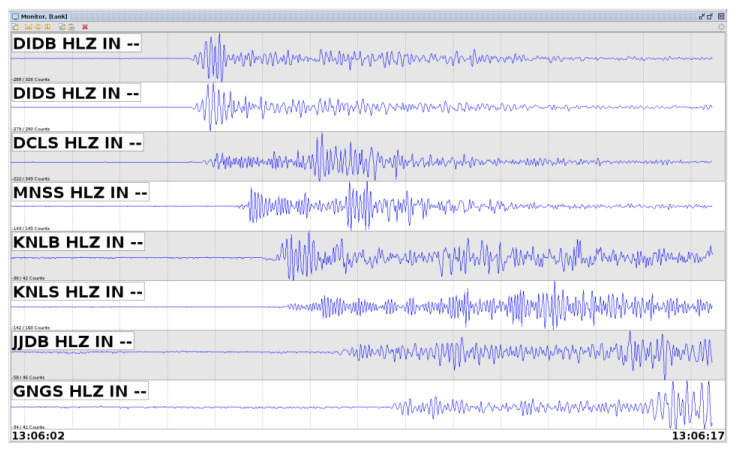
Streaming of data from sensors to the server in real-time during an earthquake event.

**Figure 4 sensors-24-03272-f004:**
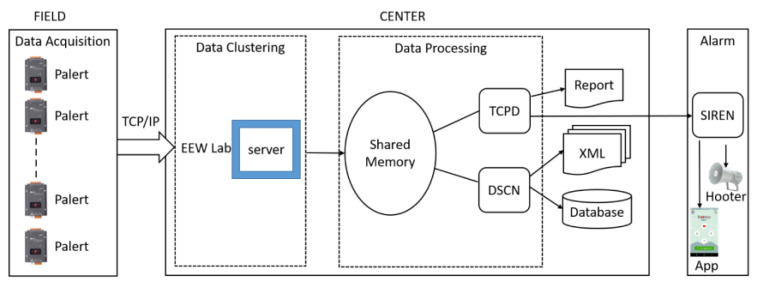
Flowchart of the UEEWS.

**Figure 5 sensors-24-03272-f005:**
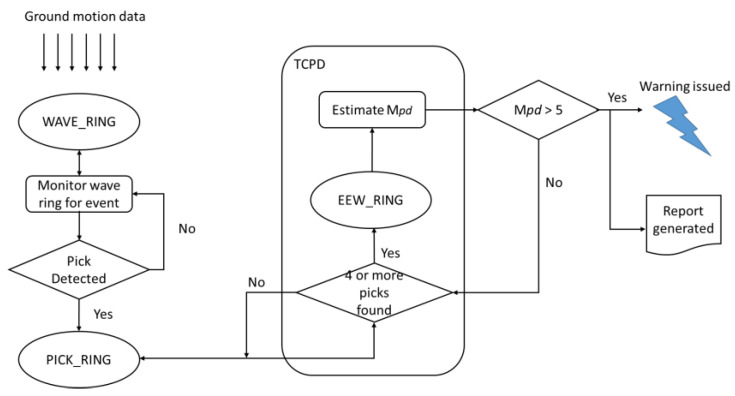
The decision-making flowchart of the UEEWS.

**Figure 6 sensors-24-03272-f006:**
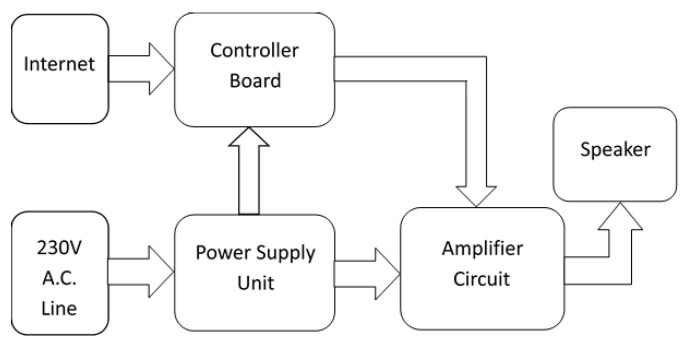
Flowchart depicting the modules of the UEEWS siren.

**Figure 7 sensors-24-03272-f007:**
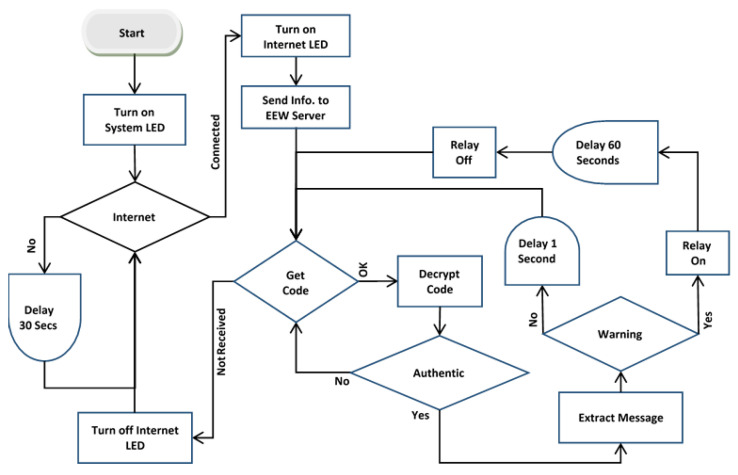
The schematic diagram for the sirens.

**Figure 8 sensors-24-03272-f008:**
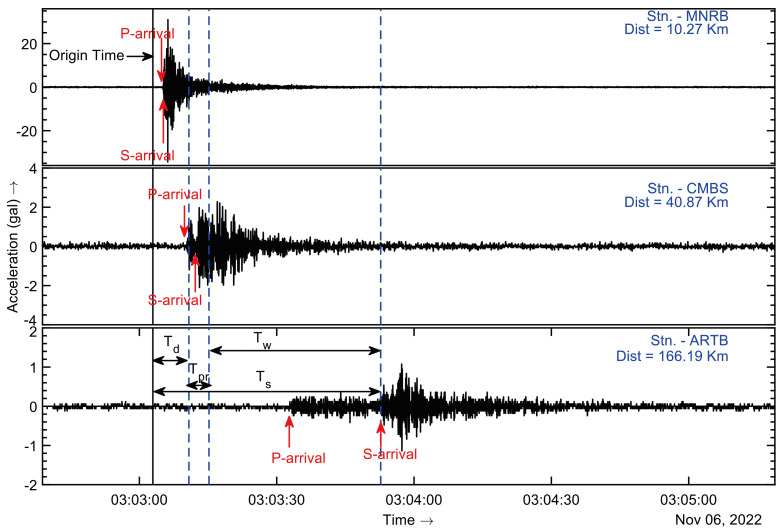
The diagram illustrates the data-recording time (*Td*), data-processing time (*Tpr*), event-reporting time (*Tr*), target-area lead time (*Tw*), and shear-wave travel time (*Ts*) during the Tehri Garhwal earthquake on 6 November 2022.

**Figure 9 sensors-24-03272-f009:**
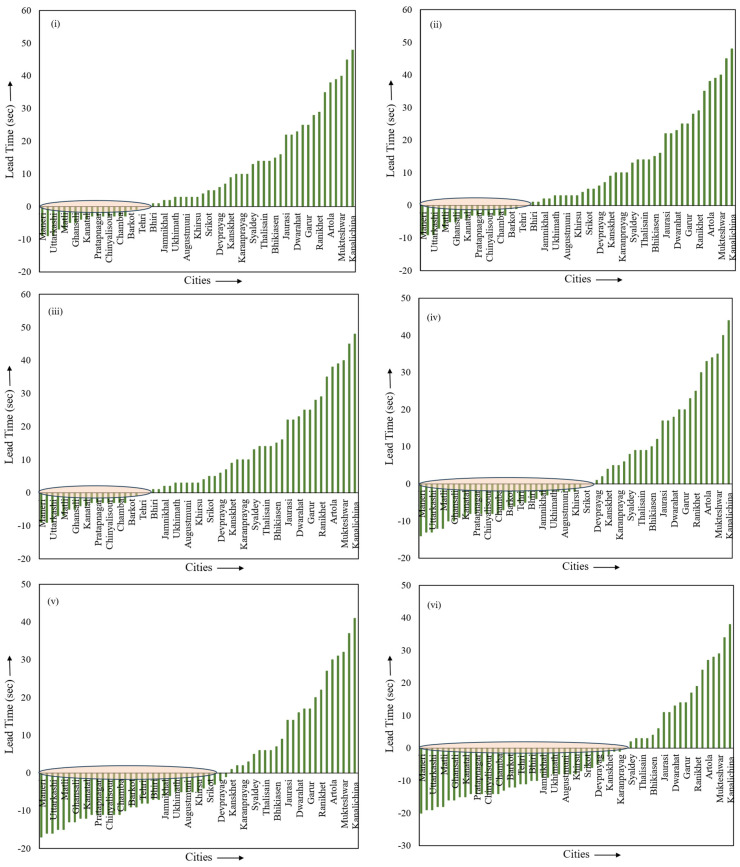
Estimated lead time for cities during the Tehri Garhwal earthquake on 6 November 2022. The plots (**i**–**vi**) represent the lead times users at different locations got during this earthquake. The pink ovals represent the blind zone.

**Figure 10 sensors-24-03272-f010:**
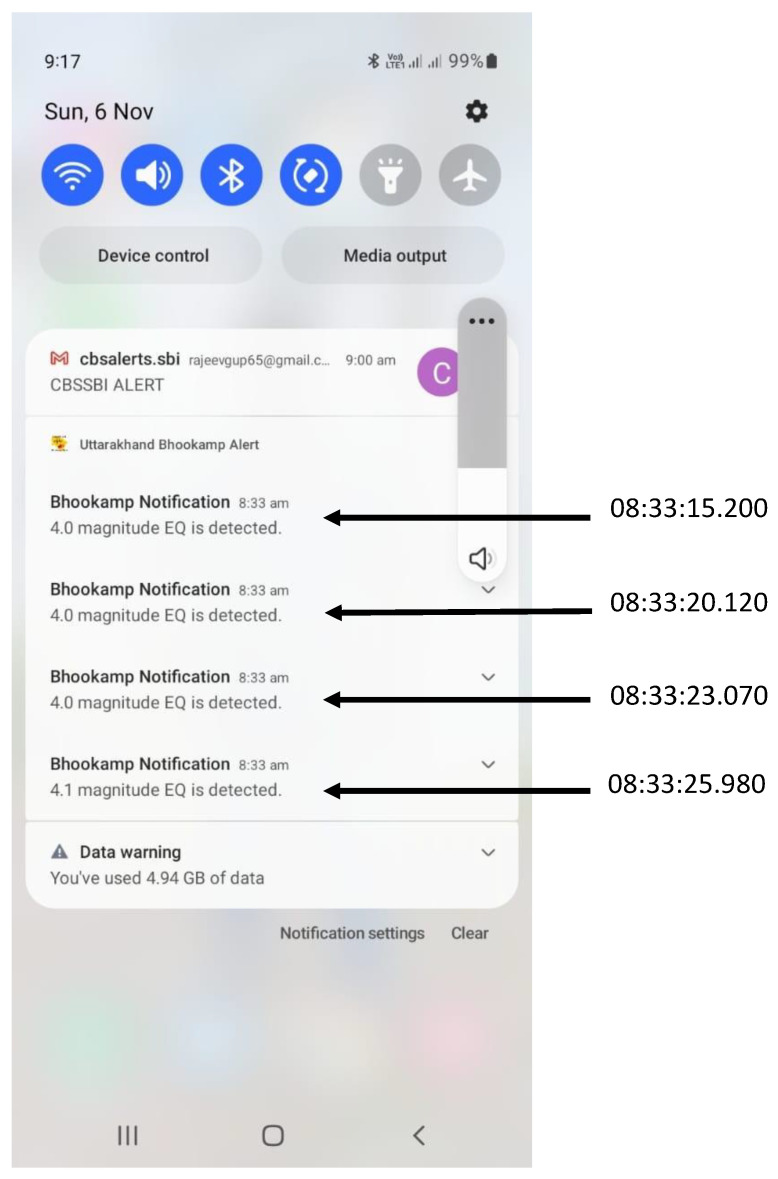
The screenshot of the notification received on the mobile app by a user.

**Figure 11 sensors-24-03272-f011:**
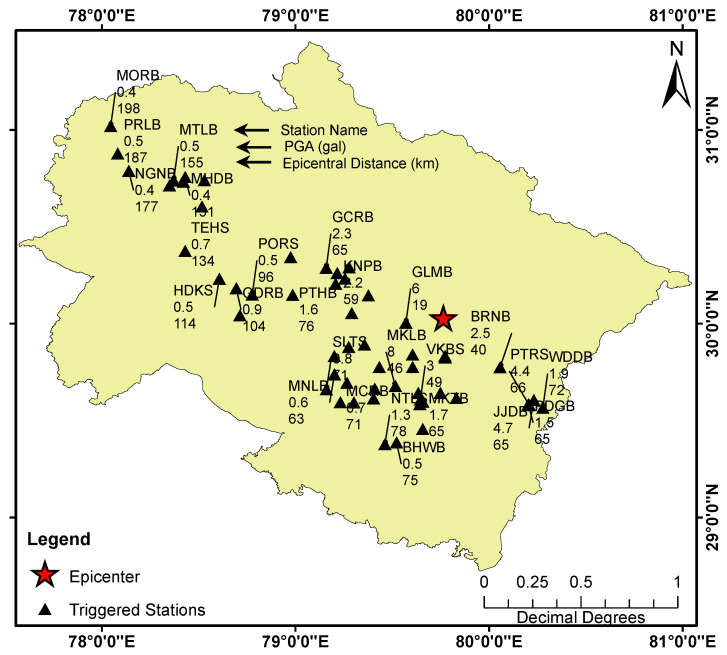
The positions of the triggered sensors along with the epicenter of the earthquake that occurred on 8 February 2020, in Pithoragarh. Each triggered sensor is labeled with three rows: the first row represents the station’s short code, the second row indicates the measured peak ground acceleration (PGA) in gal, and the third row denotes the epicentral distance in km.

**Figure 12 sensors-24-03272-f012:**
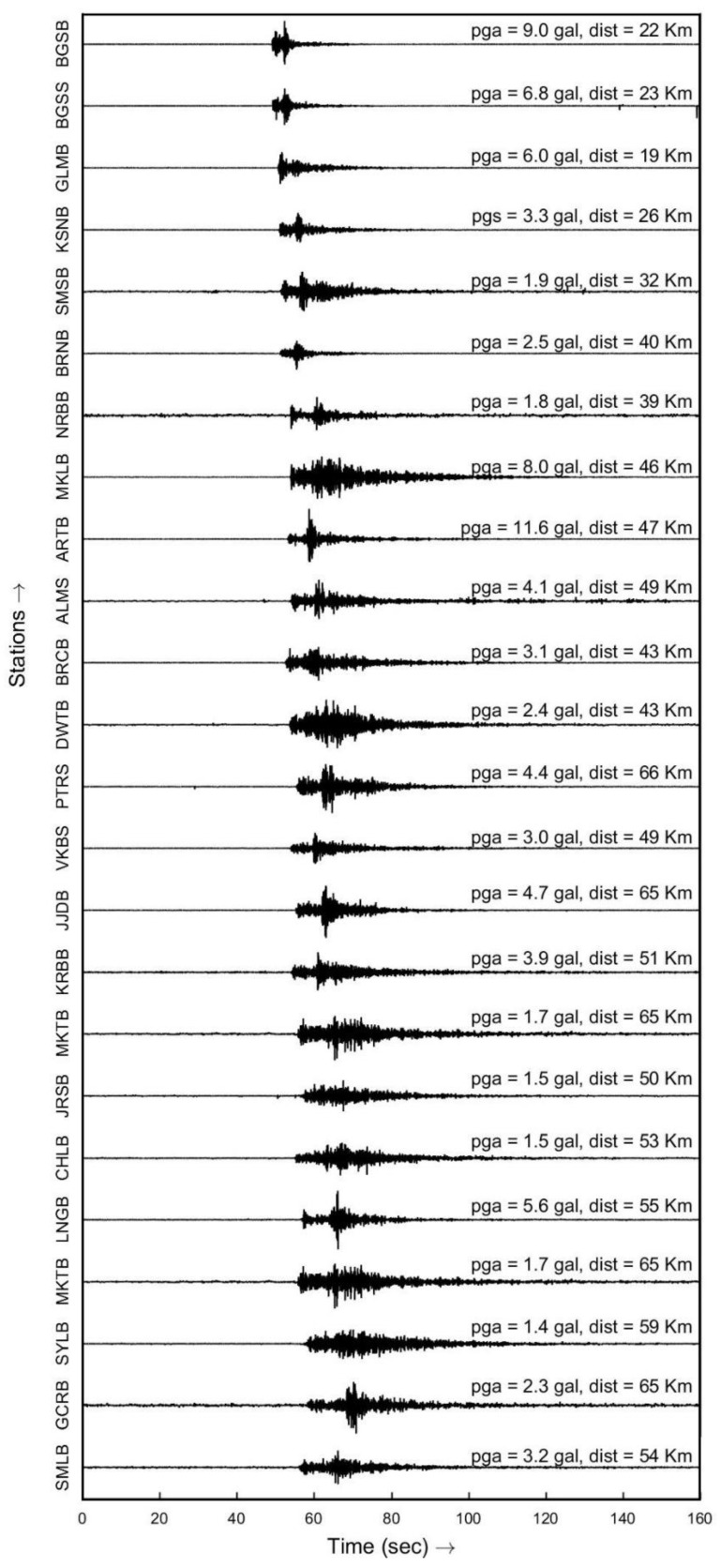
Recorded vertical channel accelerograms at different stations during the Pithoragarh earthquake on 8 February 2020.

**Figure 13 sensors-24-03272-f013:**
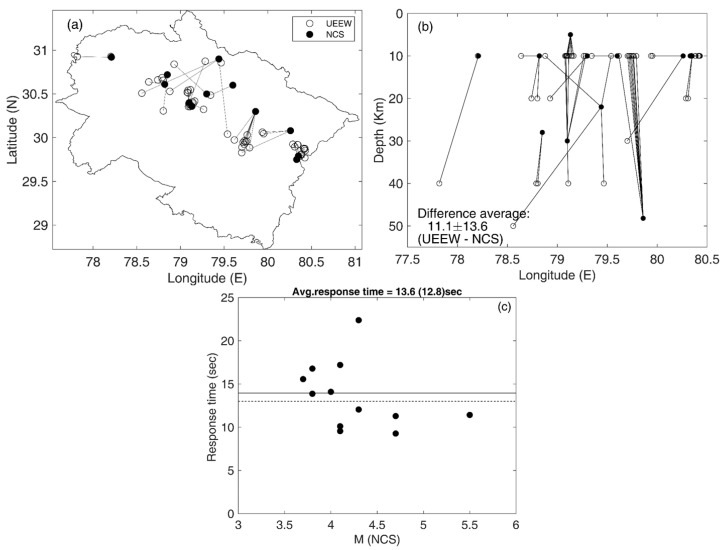
Comparison between earthquake information, including (**a**) epicenters, (**b**) depths, and (**c**) response times estimated by the UEEWS server and published on the NCS website. In (**c**), the response time is estimated based on the first reports. The dashed line in (**c**) represents the average response time i.e., 13.1 s, excluding the maximum response time of one report, which was 22.38 s.

**Figure 14 sensors-24-03272-f014:**
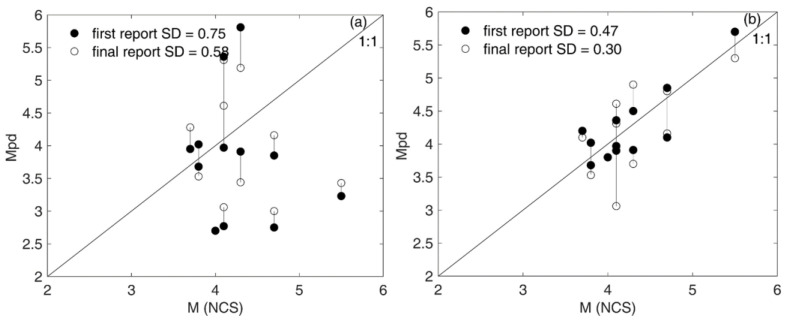
Comparison of earthquakes’ magnitude estimated by (**a**) the UEEWS in real time and NCS, and (**b**) re-running the recorded data in offline mode and NCS. The filled circles indicate the estimated magnitude in the final report, and the open circles depict the estimated magnitude in the first report.

## Data Availability

URLs of the BhuDEV mobile application are as follows: For Android users—https://play.google.com/store/apps/details?id=com.iitr.eews&pcampaignid=web_share (accessed on 28 April 2024); For iPhone users—https://apps.apple.com/in/app/bhudev/id1661902248 (accessed on 28 April 2024).

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
