# Peer review of "Uttarakhand State Earthquake Early Warning System: A Case Study of the Himalayan Environment"

_sensors, 2024, doi:10.3390/s24113272_

Round 1

Reviewer 1 Report

Comments and Suggestions for Authors

Dear authors,

The paper is interesting and relevant and it is appropriate for Sensors Journal. I recommend major revisions with comments and suggestions listed below.

The general organization of the paper is very confusing and difficult to follow. Please check the MDPI guides for authors: usually there is a Section for the Introduction (sometimes the study area is described in a separate section of the introduction), Methods, Results and Discussion (sometimes these two are separated), and Conclusions.

The objectives of the study are not clear. Please revise.

There are too many abbreviations that make the text difficult to follow. Please check if all of them are necessary; add the meaning of EEW, UrEDAS, PALERT2EW, TCPD and be consistent with the abbreviations.

Please explain in more detail the several modules (PICK_EW, WAVE_RING, etc), and add those in Figure 3. What is DSCN?

Please provide more technical details about the implementation of the early warning system.

Please renumber the order of the references. For example, [53] should be [9], and so on.

Please include the topography scale in Figure 1. Please increase the font of the schematic representations of all figures. Figure 7 must be reformulated because the locations of the sirens look very difficult to understand; the same with Figure 13. The plots of Figure 10 are complicated to interpret, please reformulate them. Figure 12 must be removed from the text because it was provided for dissemination and awareness purposes; it cannot be published in scientific papers and was not made by the authors.

Please indicate all the equations’ parameters. For example, in equation (1), what is A, Pd, C, etc? Please indicate appropriate references.

Section 2 is missing.

What do you mean by Way Forward (Section 7)?

Please check Line 242, Section 3.2. Data Streaming.

Reviewer 2 Report

Comments and Suggestions for Authors

This manuscript introduces the Uttarakhand State Earthquake Early Warning System (UEEWS), which includes 170 accelerometers and is deployed in the northern region of North Akhand, India. It issues medium to high magnitude earthquake warnings through user mobile terminal applications and alarms installed on government owned public buildings. At present, the system has successfully issued mild earthquake warnings that occur in the instrument area and moderate earthquake warnings that occur near the instrument area. This manuscript provides an overview of the design, details of the instrument, adaptation of attributes and their relationship with seismic parameters, system operation flow, and information related to alarm propagation. The content of this article is detailed, the formulas are correct, the data is reasonable, and the results are reliable.

       However, there are issues with the manuscript in the following aspects that need to be corrected.

1. There are too many keywords. There is an issue with the order of reference citation. Chapter numbering is chaotic, jumping directly from Section 1.1 to Chapter 3.

2. There is a lot of background introduction in Chapter 1, especially Section 1.1, which has a low degree of matching with the topic. It can be explained clearly in one or two sentences. Suggest reorganizing the content of Chapter 1.

3. Section 3.1 lacks a topology diagram of the collection system network.

4. Suggest changing the title of Chapter 5 to "System Application Effectiveness -A Case of the Tehri Garhwal Earthquake Warning".

5. Multiple images are unclear and the text is chaotic. There are too many pictures, some pictures are not needed, such as pictures 7 and 8.

6. The discussion, conclusion, and future outlook sections can be included in one chapter, and a systematic summary of the entire text is required.

7. The design ideas and concepts of this article are not reflected, and the description of data flow design is relatively concise. It is recommended to elaborate in the format of "data collection, preprocessing, data storage, model prediction, and final display".

8. Some English sentences are not smooth, and the author needs to optimize the structure of these English sentences.

Comments on the Quality of English Language

Some English sentences are not smooth, and the author needs to optimize the structure of these English sentences.

Round 2

Reviewer 1 Report

Comments and Suggestions for Authors

Dear authors,

Thank you very much for the cover letter; the paper has improved. Some figures continue to be difficult to interpret. 

Regarding Figure 10, If the authors want to keep it please add all the necessary methodology and citations.

Best regards.

Reviewer 2 Report

Comments and Suggestions for Authors

This revised draft has corrected some issues from the previous version, with clear objectives, detailed content, clear statements, smooth sentences, and innovative application. It is recommended to accept the revised version. The article has issues in the following areas and needs to be revised.

1. A logical structure diagram of the collection system network is missing in section 3.1.

2. Suggest placing the conclusion section before the future outlook for more logical coherence.
